# FROM LANGUAGE TO 3D WORLDS: ADAPTING LANGUAGE MODELS FOR POINT CLOUD PERCEPTION

## ABSTRACT

Typical 3D perception approaches are inclined to learn a well-performing network with supervised training on the target data or a pretraining-finetuning paradigm. Either way, they only explore in-modality solutions and data. In this work, considering that both point cloud and text are discrete data, we introduce a cross-modal strategy that applies pretrained language models for understanding 3D point clouds, where the language model is trained on language corpus and frozen. We propose a simple yet effective approach, named LAMP (LAnguage Models reading Point clouds), which merely trains a small portion of parameters to align the data distribution of 3D point clouds with pretrained language models and spark the 3D perception ability of language models. Furthermore, we utilize the 3D-aware language model to simultaneously extract features of point cloud and text, which mitigates the modality gap and boosts the performance on multimodal tasks, e.g., 3D visual grounding. Extensive experiments on unimodal and multimodal tasks validate the superiority of our proposed method.

## 1 INTRODUCTION

The point cloud is an indispensable data modality critical for numerous applications such as autonomous driving (Liu et al., 2022; Lang et al., 2019; Yin et al., 2021), robotic visual perception (Chao et al., 2021; Yang et al., 2020), and virtual reality (Xiong et al., 2021). Point cloud recognition methods typically utilize supervised learning (Qi et al., 2017b;a; Zhao et al., 2021b) or pretraining-finetuning paradigms (Pang et al., 2022; Yu et al., 2022) to facilitate model training for 3D perception tasks. These approaches, which train networks from scratch using 3D point clouds, have yielded promising results in tasks such as 3D object recognition, semantic segmentation, and part segmentation within a unimodal framework.

Specifically, a point cloud consists of a set of unordered points captured by lidar cameras, with each point encoded with spatial coordinates $x, y, z$ and, optionally, features such as intensity or color. Point clouds provide a view complementary to 2D images; while the latter captures texture and appearance, the former details an object's skeletal structure and shape. Some studies have suggested the possibility of exchanging 3D and 2D representations and adapted image-pretrained networks to encode 3D point clouds, thereby extending 3D pipelines into multimodal applications (Zhu et al., 2022; Zhang et al., 2022; Xu et al., 2022; Wang et al., 2022; Qian et al., 2022b). Nonetheless, these multimodal approaches have predominantly been explored within the vision sphere, leaving the exploration of other modalities less examined.

This work diverges from the existing literature by examining the effectiveness of non-vision modalities for 3D perception, with a focus on language. *As both texts and point clouds are discrete sets, we propose leveraging the wealth of available pretrained language models to encode 3D data*. This method circumvents the necessity of training a new model from scratch with text and point cloud data. Given the distinct nature of these modalities, we aim to adapt language models for the 3D vision domain with minimal adjustment.

In our pursuit to bridge the gap between language models and point cloud processing, we introduce a novel methodology termed LAMP, an acronym for "LAnguage Models reading Point clouds." In LAMP, the word tokenizer is substituted with the point-cloud tokenizer (Sec. 3.2) to effectively extract latent representations of raw point clouds. The subsequent challenge lies in synchronizing the

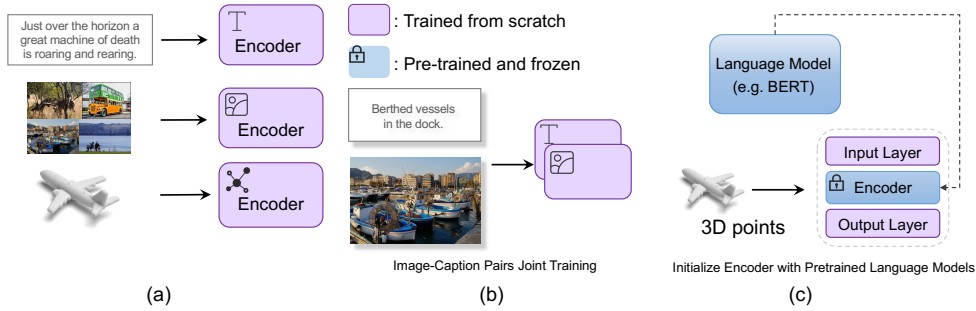

Figure 1: Our method is radically different from the previous unimodal and multimodal works. (a) The unimodal models are designed for learning representations of specific modalities. (b) The multimodal models (e.g., CLIP) trained with paired multimodal data to learn the representations by contrastive learning. (c) In this work, we directly utilize a frozen and pretrained language model to recognize 3D point clouds. Note that we do not require any language data paired with the point clouds.

encoder of language models with point clouds to engender semantically meaningful point-cloud features. To address this, we conceptualize two distinct strategies: cross-modal self-attention learning (**CMSA**) and cross-modal cross-attention learning (**CMCA**). The term "cross-modal" underscores the innovative application of parameters originally learned from natural language processing to the realm of 3D point clouds. Within the **CMSA**, each point is treated analogously to a word token and we employ self-attention mechanisms to establish inter-point correlations and facilitate information interchange. On the other hand, **CMCA** interprets the position embedding as the query and visual features as both the key and value. To align point clouds with the parameters of langauge-pretrained encoder, we utilize a lightweight projection network to transform point embeddings. It is noteworthy that both position embeddings and visual features are derived directly from raw point clouds. The inherent positional priors within the position embedding make it adept at probing visual features, thereby enhancing feature interactions. Our experimental findings underscore the efficacy of LAMP. Not only does it surpass other point-cloud methodologies that rely on specific architectural designs tailored for 3D vision, but it also showcases that a vanilla pre-trained language model can indeed interpret 3D point clouds with remarkable proficiency.

Beyond the unimodal setup, we further extend the 3D-aware language encoder to the multimodal realm. Specifically, we use one language encoder to encode both texts and point clouds in 3D visual grounding task. In this way, the modality gap between text and 3D point clouds is mitigated thanks to the shared encoding, leading to improved grounding results.

We would like to highlight that our work differs from previous works in the following aspects: 1) Compared to typical unimodal approaches, LAMP explores pretrained models from language modality to perceive point clouds, and 2) LAMP does not require paired data to align two modalities and only train a few parameters to achieve the alignment. The comparison is illustrated in Fig. 1. Extensive experiments are conducted to validate the effectiveness of our approach. For 3D object classification, LAMP achieves 93.8% OA on the ModelNet40 dataset with only 0.44M trainable parameters. For 3D semantic segmentation, LAMP outperforms other competitive methods, ACT (Dong et al., 2022) by 6.8% mIoU and Point-BERT (Yu et al., 2022) by 7.2% mIoU on the S3DIS dataset. Meanwhile, LAMP also shows an impressive performance in relieving the long-tail problem, outperforming PointNeXt (Qian et al., 2022a) by 4.3% on the tail classes of ShapeNet Part dataset (Yi et al., 2016). Furthermore, LAMP showcases strong multimodal results on ScanRefer dataset of 3D visual grounding task.

In addition, LAMP brings new directions for the vision-language area. It demonstrates that text and point clouds can be encoded by the same set of parameters.

To sum up, our contributions are as follows.

- LAMP validates the feasibility and efficacy of leveraging pretrained language models to process 3D point clouds, which opens a new avenue for 3D point cloud understanding.

- LAMP manifests that 3D point clouds and text can be encoded by the same parameters. This finding further enhances performance on text-3D tasks and 3D visual grounding.
- LAMP achieves outstanding performance on both unimodal (ModelNet-40, S3DIS, and ShapeNetPart) and cross-modal 3D visual grounding (ScanRefer), which demonstrates the effectiveness and versatility of the proposed algorithm.

## 2    RELATED WORK

### 2.1    MULTIMODAL PRETRAINING

Tranditional pretraining works developed their algorithms in the unimodal setup, such as BERT (Devlin et al., 2019), XLNet(Yang et al., 2019), and Roberta(Liu et al., 2019). Recently, several works extended pretraining approaches to the mulitomodal setup. For example, VL-BERT (Su et al., 2019) explored modality-aligned representations for generic vision-language understanding with the MLM paradigm. Oscar (Li et al., 2020) described the object semantics in both visual and textural contents. Frameworks like Vinvl (Zhang et al., 2021a), Simvlm (Wang et al., 2021c), VLMO (Wang et al., 2021b), ALBEF (Li et al., 2021), and Florence (Yuan et al., 2021a) further develop the joint representations across vision-language modalities in terms of semantic consistency. Different from these works, the proposed LAMP aims to directly adapt a language-pretrained model to 3D modality.

### 2.2    POINT-ORIENTED NETWORKS

In the early stage, convolutional layers are leveraged to construct the network to encode 3D point clouds. Usually, researchers set abstraction of original input points and utilize grids for precise and regular representation (Qi et al., 2017b;a; Zhao et al., 2019; Thomas et al., 2019). Also, to exploit the advantages of convolutional networks, they design different image planes and employ CNN backbones to extract representations of points (Su et al., 2015; Chen et al., 2017; Lang et al., 2019). Meanwhile, the transformation between irregular points and regular voxel also brings improvements (Maturana & Scherer, 2015; Song et al., 2017), which also depends on CNN for feature extraction. (Graham et al., 2018) and (Choy et al., 2019) further enhanced voxel-based methods by proposing sparse convolution to improve computational efficiency. Recently, with the advancement of transformer architecture (Vaswani et al., 2017), (Zhao et al., 2021a) and (Guo et al., 2021) both introduce the attention mechanism to point cloud understanding. (Park et al., 2022) proposed a hash-based scheme and lightweight attention layers for point transformer architecture (Zhao et al., 2021a), which successfully boosts the efficiency. In addition, pre-training is also a trending means to enhance perception performance on 3D point clouds (Yu et al., 2022). PointContrast (Xie et al., 2020) learns scene level representation via contrastive learning across different views of a point cloud. (Zhang et al., 2021b) further extended pretraining paradigm to single-view 3D data. (Hou et al., 2021) enhanced PointContrastive by integrating spatial information into contrastive learning. MSC (Wu et al., 2023a) and PointClustering (Long et al., 2023) introduce reconstructive learning and clustering as the pretext task of pretraining. In comparison, our work explore a cross-modal strategy instead of focusing on use point-cloud data to train a well-performed network from scratch.

### 2.3    CROSS-MODAL KNOWLEDGE TRANSFER TO POINT

Recently, an increased number of works focused on transferring learned knowledge from other modalities to 3D point clouds. Image2Point (Xu et al., 2022; Wang et al., 2022; Qian et al., 2022b) found that vision-pretrained models can be adapted to perceive point clouds. After witnessing the success in 2D zero-shot and few-shot learning of CLIP (Radford et al., 2021), several methods align point clouds with CLIP-style models, e.g. CLIP and GLIP (Li et al., 2022), to fulfill open-world 3D recognition (Zhang et al., 2022; Peng et al., 2023; Rozenberszki et al., 2022; Zhang et al., 2023; Xue et al., 2023; Liu et al., 2023b;a). Some other works build a strong pertaining 3D baseline model with CLIP (Xue et al., 2023; Peng et al., 2023; Liu et al., 2023a) or a caption model (Ding et al., 2022). Different from LLaMa AdapterV2 (Gao et al., 2023) and Point-Bind & Point-LLM (Guo et al., 2023), LAMP excels in handling long-tailed and out-of-distribution problems in point cloud analysis and joint multimodal learning in the 3D visual grounding tasks.

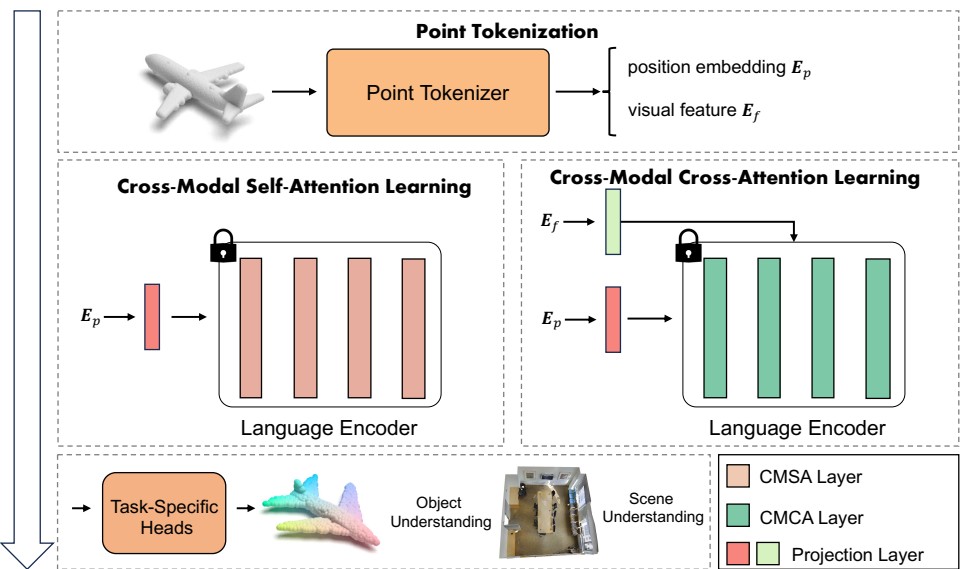

Figure 2: **Framework of LAMP**. It contains three parts: point tokenizer, encoder, and task-specific heads. Point tokenizer projects point clouds from input space to latent space, resulting in high-dimensional numeric vectors which are also referred to as point embeddings. Point tokenizer will generate two kinds of point embeddings: position embedding and visual features. We design two schemes for transferring language encoders: cross-modal self-attention learning and cross-modal cross-attention learning. Both schemes fix the langauge-pretrained encoder and only train lightweight projection layers. Finally, the task-specific head outputs predictions for downstream tasks such as object understanding and scene understanding.

Nevertheless, existing approaches all follow the "consensus" that neural networks could obtain the ability to deal with the modalities only after they got trained with the corresponding data, which indicates that the ability for modality generalization of neural networks has not been explored. In this paper, we utilize networks only pretrained with corpus and tune networks on the 3D point cloud understanding. LAMP is the first general vision-language framework, which intuitively utilizes the learned patterns from natural language to transfer knowledge to 3D point clouds.

## 3 Language Model Read Points

### 3.1 Revisit Language Transformer Models

At the core of LAMP is perceiving 3D point clouds with language-pretrained transformer models. Transformer-based language models (Devlin et al., 2019; Liu et al., 2019) follow an encoder-decoder architecture with self-attention (Vaswani et al., 2017) and have achieved state-of-the-art performance on various NLP tasks. There are 3 steps for transformer models to extract representations from natural language. 1) Word Tokenization. A word token is a sequence of characters that are grouped together as a useful semantic unit for processing. Referring to the vocabulary (Wu et al., 2016) (the set of all unique tokens), a sentence can be tokenized into a set of word tokens including special tokens, `SEP` and `CLS`, which indicate the separator and representation of entire input sequence, respectively. 2) Token to Embedding: The embedding of each word token is learned with embedding layers of a neural network, which projects each token to high-dimensional vector space. 3) Sequence Modeling: To inject the relative or absolute position of the tokens in the sentence, position embeddings are further added to the word embeddings to jointly represent the original sentence. These embeddings are fed into sequence-to-sequence models to output the contextualized representation.

In this paper, we leverage the weights of language transformer models to encode point clouds, transferring the knowledge learned from sequences of word tokens to uncover the structure and features of point clouds. A critical problem would be how to facilitate the adaptation of language models to 3D vision.

First of all, the word tokenizer is obviously not adequate to extract latent representation from raw points, therefore we substitute the word tokenizer with a point tokenizer with aligned output dimension as described in Sec. 3.2. Then, we design two schemes to facilitate the parameters of the language encoder to understand point clouds, which are cross-modal self-attention learning in Sec. 3.3 and cross-modal cross-attention learning in Sec. 3.3. The design of task-specific heads follows the common-used solution.

## 3.2 POINT TOKENIZATION

To employ the language encoder to process 3D points, the first step is to transform them from input space to high-dimensional vector space. Formally, denote a point cloud of $N$ points as $\mathcal{X} = \{\boldsymbol{x}_i\}_{i=1}^{N}$, with $\boldsymbol{x}_i = (\boldsymbol{p}_i, \boldsymbol{f}_i)$, where $\boldsymbol{p}_i \in \mathbb{R}^3$ is the 3D location and $\boldsymbol{f}_i \in \mathbb{R}^c$ is the feature of the $i$-th point. In general, $\boldsymbol{f}_i$ contains the visual information regarding color, viewpoint, normal, *etc*. Similar to Pix4Point (Qian et al., 2022b), we utilize a trainable point tokenizer to extract embeddings of the point cloud. Specifically, we utilize the Farthest Point Sampling (FPS) operation to sample a subset of points with a fixed sampling rate (1/4), and then use convolution to extract features of the sub-clouds. Such a two-step transformation can be summarized as

$$\mathcal{X} \in \mathbb{R}^{N \times (3+c)} \rightarrow \mathcal{X}' \in \mathbb{R}^{\frac{N}{4} \times \frac{c'}{2}} \rightarrow \mathcal{X}'' \in \mathbb{R}^{\frac{N}{16} \times c'}, \tag{1}$$

where $c'$ is the dimension of resultant point embedding. In each stage, we employ FPS operation and $k$-nearest neighbor to obtain subsets of current sets, which is similar to segmenting sentences into token sequences. After the two-step transformation, we utilize Batch Normalization (BN), max pooling (max), and ReLU activation function to obtain 3D position embeddings as

$$\boldsymbol{p}''_{m,n} = \text{MLP}(\text{BN}([\boldsymbol{f}'_{m,n} - \boldsymbol{f}'_m, \boldsymbol{p}'_{m,n} - \boldsymbol{p}'_m])), \quad \boldsymbol{E}_p = \text{MLP}'(\text{ReLU}([\boldsymbol{p}''_{m,n}, \max_{n:(m,n)\in\mathcal{N}} \boldsymbol{p}'_{m,n}])), \tag{2}$$

where $\boldsymbol{p}'_{m,n}$ and $\boldsymbol{f}'_{m,n}$ are the 3D coordinates and features of $n$-th neighbor of $m$-th center point sampled by FPS, and $p'_m$ denotes the center. To obtain visual feature embeddings $\boldsymbol{E}_f$, we take a similar operation with feature $\boldsymbol{f}$ as 3D position coordinates. In summary, after point tokenization, we deal with position and feature input and obtain 3D position and visual feature embeddings $\boldsymbol{E}_p$, $\boldsymbol{E}_f \in \mathbb{R}^{N \times L \times C}$, where $L$ denotes the length of sequences and $C$ denotes the embedding dimension.

## 3.3 POINT-LANGUAGE ENCODER

**Cross-Modal Self-Attention Learning (CMSA)**. In the original language encoder, self-attention is utilized to exchange the information of each word token. After layer-by-layer interaction among word tokens, the encoder generates high-level semantic features of input language sequences. Cross-modal self-attention aims to effectively transfer the above process to 3D point clouds. To achieve this, we prepend an MLP project network before the encoder. The role of the projector is to align point-cloud data with the parameters of the language encoder.

Formally, let $f$ denote the projector, we feed position embedding of point clouds into $f$ and obtain corresponding outputs: $\mathbf{z}_p^0 = F_p(\mathbf{E}_p)$, which is the input of subsequent encoder. $\mathbf{W}_q^L, \mathbf{W}_k^L, \mathbf{W}_v^L$ ($L$ stands for that weights are language-pretrained) denote the linear layers for queries, keys, and values in language encoder, then we can formulate the cross-modal self-attention as:

$$z_p^i = \frac{\mathbf{W}_q^{L,i}\mathbf{z}_p^{(i-1)}\mathbf{W}_k^{L,i}\mathbf{z}_p^{(i-1)}}{\sqrt{d}}\mathbf{W}_v^{L,i}\mathbf{z}_p^{(i-1)}, \quad s.t.\ i = 1, 2, ..., n, \tag{3}$$

where $i$ indicates i$^{th}$ layer of encoder and $d$ is the dimension of query embedding.

**Cross-Modal Cross-Attention Learning (CMCA)**. CMSA only utilizes the position information of point clouds. Hence, we further introduce CMCA to leverage position information and visual features at the same time. The key design of CMCA is to cast the position embedding as a query

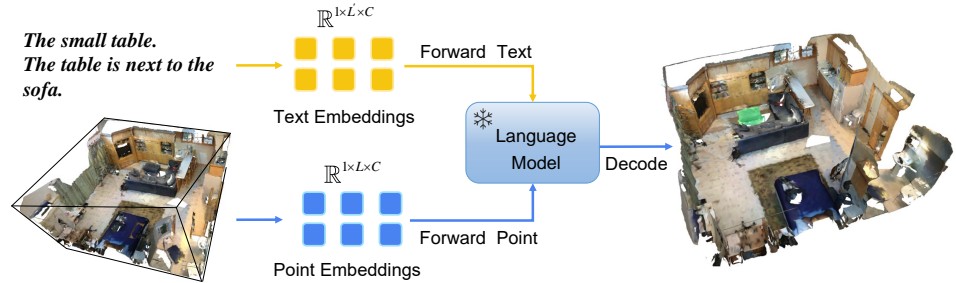

Figure 3: In this paper, we utilize networks only pretrained with corpus and tune networks on the 3D point cloud understanding. LAMP is the first general vision-language framework, which intuitively utilizes the learned patterns from natural language to transfer knowledge to 3D point clouds.

since it carries spatial prior and cast the visual feature as the key and value since it carries rich semantic information. Similar to CMSA, we leverage two MLP networks $F_p, F_f$ to align position embeddings and visual features with language encoder, i.e., $\mathbf{z}_p = F_p(\mathbf{E}_p), \mathbf{z}_f = F_f(\mathbf{E}_f)$.

Technically, we define CMCA as

$$z_p^i = \frac{\mathbf{W}_q^{L,i}\mathbf{z}_p^{(i-1)}\mathbf{W}_k^{L,i}\mathbf{z}_f^{(i-1)}}{\sqrt{d}}\mathbf{W}_v^{L,i}\mathbf{z}_f^{(i-1)}, \quad s.t.\ i = 1, 2, ..., n. \quad (4)$$

We highlight the difference between Eq. 4 and Eq. 3 with blue color.

**Optimization procedure**. In our architecture, we substitute original attention layers in language encoders with our proposed CMSA and CMSA layers, respectively. In both schemes, only the project networks $F_p, F_j$ are updated, while the parameters of language-pretrained encoder are frozen during training. A task-specific head is attached after the encoder to perform task predictions, which is also updated.

**Extending to 3D visual grounding**. We further extend LAMP to a multimodal scenario, i.e., 3D visual grounding, where a text is given to locate the corresponding object in a 3D scene. LAMP unifies the encoders for perceiving different input modalities with a single language model. The language encoder is used to simultaneously encode point clouds and texts. Thanks to the unified encoding, the modality gap between input modalities is reduced, leading to better grounding performance.

## 4 EXPERIMENTS

We first describe the experiment setup including dataset information and implementation details, followed by the comparison results with other competitive results on both unimodal and cross-modal benchmarks. Then, we show the analytical experiments for LAMP in terms of several factors. Furthermore, we discuss the performance of LAMP under long-tailed and domain shift settings.

### 4.1 EXPERIMENT SETUP

**Unimodal**: We evaluate LAMP on several datasets across classification and segmentation tasks. 1) **Classification**: To validate the performance of LAMP on the 3D object classification, ModelNet-40 (Wu et al., 2015) is utilized as the benchmark. It contains CAD models of 40 classes with 9,843 samples for training and 2,468 samples for validation. 2) **Semantic segmentation**: We evaluate LAMP on both S3DIS (Armeni et al., 2016) and ShapeNetPart (Yi et al., 2016) datasets. *S3DIS* covers 6 large indoor areas and 13 semantic classes, comprising 271 rooms. There are 16, 880 models in 16 different shape categories in the ShapeNetPart dataset. Particularly, the head for segmentation consists of a three-layer convolution network with output dimensions depending on the class number of datasets. For all experiments, there are several default hyper-parameters in the neural networks: we split $N_g = 32$ groups to preprocess points and we set the dropout rate $p = 0.1$ for attention layers. To optimize our network, we employ cosine learning rate scheduler (Loshchilov & Hutter, 2016) and AdamW optimizer (Loshchilov & Hutter, 2019) with a weight decay of 0.05.

Table 1: **Comparisons among design choices** on ModelNet-40. We report overall accuracy (OA) and mean accuracy scores (mAcc) (%). In the default setting, which is labeled in gray, the encoder has 12 transformer blocks and 6 heads.

(a) Pretrained Corpus.

| Models | State | OA | mAcc |
|---|---|---|---|
| BERT | English | 91.41 | 88.18 |
| BERT | Chinese | 90.64 | 87.02 |
| BERT | Multilingual | 91.17 | 87.28 |

(b) Model Scale.

| Model | Size | OA | mAcc |
|---|---|---|---|
| BERT | small | 90.35 | 87.24 |
| BERT | Base | 91.90 | 88.28 |
| BERT | Large | 91.49 | 88.18 |

(c) Model architecture.

| Models | State | OA | mAcc |
|---|---|---|---|
| Roberta | Finetune | 91.65 | 88.18 |
| T5 | Finetune | 92.63 | 88.28 |
| XLNet | Finetune | 89.22 | 84.29 |

(d) Pretraining text case.

| Models | State | OA | mAcc |
|---|---|---|---|
| BERT | Uncased | 91.41 | 88.18 |
| BERT | Cased | 87.20 | 82.10 |
| XLNet | Cased | 88.54 | 83.60 |
| XLNet | Large+Cased | 85.14 | 82.03 |

(e) Parameter Frozen.

| Models | State | OA | mAcc |
|---|---|---|---|
| BERT | Frozen | 90.84 | 86.84 |
| Roberta | Frozen | 89.71 | 85.08 |
| T5 | Frozen | 92.79 | 89.78 |
| XLNet | Frozen | 88.33 | 83.52 |

(f) Learning Schemes.

| Models | Approach | OA | mAcc |
|---|---|---|---|
| BERT | CMCA | 89.10 | 84.53 |
| Roberta | CMCA | 86.51 | 80.82 |
| T5 | CMCA | 91.86 | 88.65 |
| XLNet | CMCA | 87.64 | 83.29 |

**Cross-modal**: We evaluate LAMP on a cross-modal task, 3D visual grounding. ScanRefer dataset is widely adopted for this task, which comprises visual data of ScanNet (Dai et al., 2017) and 51, 538 text descriptions. We use the point tokenizer (Sec. 3.2) and Roberta's word tokenizer for tokenization and Roberta model (Liu et al., 2019) to simultaneously extract the semantic features of point clouds and texts and fuse such multimodal features. Then the fused features are fed into a 6-layer transformer-based decoder to predict bounding boxes and labels. It is worth noting that the shared language model is frozen.

Table 2: **3D Object Classification on ModelNet-40.** We report the pre-training modality (Pre-train) and **trainable** parameters number (Param).

| Method | Pre-train | mAcc (%) | OA (%) | Params |
|---|---|---|---|---|
| PointNet [CVPR'17] (Qi et al., 2017b) | N/A | 86.0 | 89.2 | 3.5M |
| PointNet++ [NeurIPS'17] (Qi et al., 2017a) | N/A | - | 91.9 | 1.5M |
| PointCNN [NeurIPS'18] (Li et al., 2018) | N/A | 88.1 | 92.5 | 0.6M |
| PointConv [CVPR'19] (Wu et al., 2019) | N/A | - | 92.5 | - |
| KPConv [ICCV'19] (Thomas et al., 2019) | N/A | - | 92.9 | 14.3M |
| DGCNN [TOG'19] (Wang et al., 2019b) | N/A | 90.2 | 92.9 | 1.8M |
| Point Transformer [ICCV'21] (Zhao et al., 2021b) | N/A | 90.6 | 93.7 | 7.8M |
| PointNeXt [NeurIPS'22] (Qian et al., 2022a) | N/A | 90.8 | 93.2 | 1.4M |
| Point-MLP [ICLR'22] (Ma et al., 2022b) | N/A | 90.9 | 93.6 | 0.68M |
| PointMixer [ECCV'22] (Choe et al., 2022) | N/A | 91.4 | 93.6 | 3.6M |
| Point-BERT [CVPR'22] (Yu et al., 2022) | 3D | - | 93.2 | 21.1M |
| Point-MAE [ECCV'22] (Pang et al., 2022) | 3D | - | **93.8** | 21.1M |
| P2P [NeurIPS'22] (Wang et al., 2022) | 2D | - | 93.1 | 1.2M |
| ACT [ICLR'23] (Dong et al., 2022) | 2D | - | 93.5 | 21.1M |
| LAMP (Ours) | Language | 90.1 | **93.8** | **0.44M** |

## 4.2 Comparisons among Design Choices

We experiment to evaluate multiple design choices such as pretraining corpus, model scale, architecture, *etc*. Table 1 presents the results of point cloud classification on ModelNet-40.

**Pretrained Corpus**. We compare the point cloud understanding capability of language models pretrained on different text corpora and the results are shown in Table 1a.

**Model Size**. We further examine whether larger language models could improve point cloud understanding capability. Table 1b shows that a larger language model does not necessarily achieve better point cloud recognition performance on ModelNet-40.

**Model architecture**. Table 1c shows the results with different language models including Roberta (Liu et al., 2019), T5 (Raffel et al., 2020), and XLNet (Yang et al., 2019).

**Text Case**. From Table 1d, where "Cased" means that uppercase and lowercase words represent different semantics, we can infer that a language model not pretrained to be discriminative to text cases could lead to better point cloud recognition performance.

**Parameters Frozen**. In contrast to tuning language models, we propose to freeze the language model parameters. Table 1e shows that language model can directly encode point clouds.

**Point Language Encoder**. We conduct experiments with the encoder learning scheme (Sec. 3.3), with results shown in Table 1f. It reveals that a language model is also a point cloud encoder.

## 4.3 Results on Shape Classification

The LAMP model's performance on the ModelNet-40 dataset is detailed in Table 1, outperforming traditional point cloud methods such as PointNet++(Qi et al., 2017a), PointCNN(Li et al., 2018), and

Point Transformer (Zhao et al., 2021b). It also excels against multimodal methods like P2P (Wang et al., 2022), which starts with image pretraining, and ACT (Dong et al., 2022), which utilizes image and language data, demonstrating LAMP's superior classification capabilities.

Table 3: **Part segmentation results on the ShapeNetPart dataset**. We report the mean IoU across all part categories mIoU$_C$ (%), the mean IoU across all instances mIoU$_I$ (%), and the IoU (%) for each category. The best and second best experimental results are **bolded** and underlined respectively.

| Model | mIoU$_C$ | mIoU$_I$ | aero plane | bag | cap | car | chair | ear phone | guitar | knife | lamp | laptop | motor bike | mug | pistol | rocket | skate board | table |
|---|---|---|---|---|---|---|---|---|---|---|---|---|---|---|---|---|---|---|
| PointNet [CVPR'17] (Qi et al., 2017b) | 80.4 | 83.7 | 83.4 | 78.7 | 82.5 | 74.9 | 89.6 | 73.0 | 91.5 | 85.9 | 80.8 | 95.3 | 65.2 | 93.0 | 81.2 | 57.9 | 72.8 | 80.6 |
| PointNet++ [NeurIPS'17] (Qi et al., 2017a) | 81.9 | 85.1 | 82.4 | 79.0 | 87.7 | 77.3 | 90.8 | 71.8 | 91.0 | 85.9 | 83.7 | 95.3 | 71.6 | 94.1 | 81.3 | 58.7 | 76.4 | 82.6 |
| DGCNN [TOG'19] (Wang et al., 2019a) | 82.3 | 85.2 | 84.0 | 83.4 | 86.7 | 77.8 | 90.6 | 74.7 | 91.2 | 87.5 | 82.8 | 95.7 | 66.3 | 94.9 | 81.1 | 63.5 | 74.5 | 82.6 |
| KPConv [ICCV'19] (Thomas et al., 2019) | 85.1 | 86.4 | 84.6 | 86.3 | 87.2 | 81.1 | 91.1 | 77.8 | **92.6** | **88.4** | 82.7 | 96.2 | 78.1 | 95.8 | **85.4** | **69.0** | 82.0 | 83.6 |
| OcCo [ICCV'21] (Wang et al., 2021a) | 83.4 | 85.1 | 83.3 | 85.2 | 88.3 | 79.9 | 90.7 | 74.1 | 91.9 | 87.6 | 84.7 | 95.4 | 75.5 | 94.4 | 84.1 | 63.1 | 75.7 | 80.8 |
| Point-BERT [CVPR'22] (Yu et al., 2022) | 84.1 | 85.6 | 84.3 | 84.8 | 88.0 | 79.8 | 91.0 | 81.7 | 91.6 | 87.9 | 85.2 | 95.6 | 75.6 | 94.7 | 84.3 | 63.4 | 76.3 | 81.5 |
| Point-MLP [ICLR'22] (Ma et al., 2022a) | 84.6 | 86.1 | 83.5 | 83.4 | 87.5 | 80.5 | 90.3 | 78.2 | 92.2 | 88.1 | 82.6 | 96.2 | 77.5 | 95.8 | **85.4** | 64.6 | **83.3** | **84.3** |
| Point-MAE [ECCV'22] (Pang et al., 2022) | 84.2 | 86.1 | 84.3 | 85.0 | 88.3 | 80.5 | 91.3 | 78.5 | 92.1 | 87.4 | **86.1** | 96.1 | 75.2 | 94.6 | 84.7 | 63.5 | 77.1 | 82.4 |
| P2P [NeurIPS'22] (Wang et al., 2022) | 84.1 | 86.5 | 84.3 | 85.1 | 88.3 | 80.4 | 91.6 | 80.8 | 92.1 | 87.9 | 85.6 | 95.9 | 76.1 | 94.2 | 82.4 | 62.7 | 74.7 | 83.7 |
| PointNeXt [NeurIPS'22] (Qian et al., 2022a) | 84.2 | 86.7 | 85.2 | 84.7 | 85.3 | 81.5 | 91.8 | 79.3 | 91.8 | 87.9 | 85.1 | 96.1 | 75.5 | 95.9 | 83.6 | 62.9 | 76.4 | 83.8 |
| Pix4Point [Arxiv'22] (Qian et al., 2022b) | 84.1 | 86.5 | 85.7 | 87.5 | 87.0 | 81.8 | 92.0 | 83.9 | 92.6 | 88.8 | 85.2 | 96.2 | **80.3** | 96.1 | 84.9 | 65.5 | 78.1 | 82.8 |
| LAMP (Ours) | **85.4** | **86.9** | 85.8 | **89.5** | **89.5** | **82.2** | **91.9** | **83.3** | 92.5 | 87.9 | 85.2 | **96.3** | 80.2 | **96.2** | 85.2 | 58.2 | 80.2 | 82.9 |

## 4.4 RESULTS ON SEMANTIC SEGMENTATION

Table 3 shows LAMP outperforms existing advanced models including Point-BERT (Yu et al., 2022), Point-MLP (Ma et al., 2022a), and Point-MAE (Pang et al., 2022) in terms of both mIoU$_C$ and mIoU$_I$. Meanwhile, a new state-of-the-art Instance mIoU of 86.9% is achieved on the ShapeNetPart dataset. Except for mIoU, the respective IoU scores of many categories are also impressive.

Table 4: Out-of-domain recognition (accuracy %) on the PointDA-10 dataset. M: ModelNet, S: ShapNet, S*: ScanNet; $\rightarrow$ indicates the adaptation direction. Adv.: adversarial domain alignment, SLT: self-learning tasks, and SPST: self-paced self-training.

| Methods | Adv. | SLT | SPST | M $\rightarrow$ S | M $\rightarrow$ S* | S $\rightarrow$ M | S $\rightarrow$ S* | S* $\rightarrow$ M | S $\rightarrow$ S | Avg. |
|---|---|---|---|---|---|---|---|---|---|---|
| DANN [JMLR'16] (Ganin et al., 2016) | ✓ | | | 74.8 ± 2.8 | 42.1 ± 0.6 | 57.5 ± 0.4 | 50.9 ± 1.0 | 43.7 ± 2.9 | 71.6 ± 1.0 | 56.8 |
| PointDAN [NeurIPS'19] (Qin et al., 2019) | ✓ | | | 83.9 ± 0.3 | 44.8 ± 1.4 | 63.3 ± 1.1 | 45.7 ± 0.7 | 43.6 ± 2.0 | 56.4 ± 1.5 | 56.3 |
| RS [NeurIPS'19] (Sauder & Sievers, 2019) | | ✓ | | 79.9 ± 0.8 | 46.7 ± 4.8 | 75.2 ± 2.0 | 51.4 ± 3.9 | 71.8 ± 2.3 | 71.2 ± 2.8 | 66.0 |
| DefRec [CVPR'21] (Achituve et al., 2021) | | ✓ | | 81.7 ± 0.6 | 51.8 ± 0.3 | 78.6 ± 0.7 | 54.5 ± 0.3 | 73.7 ± 1.6 | 71.1 ± 1.4 | 68.6 |
| GAST [CVPR'21] (Zou et al., 2021) | | ✓ | | 83.9 ± 0.2 | 56.7 ± 0.3 | 76.4 ± 0.2 | 55.0 ± 0.2 | 73.4 ± 0.3 | 72.2 ± 0.2 | 69.5 |
| | | ✓ | ✓ | 84.8 ± 0.1 | **59.8 ± 0.2** | 80.8 ± 0.6 | 56.7 ± 0.2 | 81.1 ± 0.8 | 74.9 ± 0.5 | 73.0 |
| Implicit PCDA [CVPR'22] (Shen et al., 2022) | | ✓ | | 85.8 ± 0.3 | 55.3 ± 0.3 | 77.2 ± 0.4 | 55.4 ± 0.5 | 73.8 ± 0.6 | 72.4 ± 1.0 | 70.0 |
| | | | ✓ | 86.2 ± 0.2 | 58.6 ± 0.1 | **81.4 ± 0.4** | 56.9 ± 0.2 | **81.5 ± 0.5** | 74.4 ± 0.6 | **73.2** |
| MLSP [ECCV'22] (Liang et al., 2022) | | ✓ | | 83.7 ± 0.4 | 55.4 ± 1.8 | 77.1 ± 0.9 | 55.6 ± 0.7 | 78.2 ± 1.5 | 76.1 ± 0.5 | 71.0 |
| | | | ✓ | 85.7 ± 0.6 | 59.4 ± 1.3 | 82.3 ± 0.9 | 57.3 ± 0.7 | **82.2 ± 0.5** | **76.4 ± 0.5** | 73.8 |
| LAMP [ours] | | ✓ | | **86.2 ± 0.8** | 59.1 ± 0.9 | **83.5 ± 0.4** | **57.6 ± 0.6** | 81.2 ± 0.4 | **76.4 ± 0.3** | **74.0** |

## 4.5 LONG-TAILED AND OUT-OF-DOMAIN RECOGNITION IN 3D VISION TASKS

**Long-tailed recognition**. We explore how language models perform under the long-tailed setup, and evaluate long-tailed performance on ShapeNetPart (Yi et al., 2016) dataset. As shown in Table 6, we divide original ShapeNetPart categories to 3 groups: Many, Medium, and Few according to the number of training samples. We calculate the mean IoU scores for the three types. Compared with existing advanced models such as KPConv (Thomas et al., 2019), Point-MAE (Pang et al., 2022), and Point-MLP (Ma et al., 2022a), LAMP achieves better performance, especially for the tail classes. In LAMP, the language encoder is pretrained on the language corpus and frozen during training, which prevents it from biasing by the long-tailed distribution. Thus, LAMP can alleviate long-tailed performance to some extent.

**Out-of-domain (ODD) recognition**. We compare LAMP with other methods on the out-of-domain benchmark, i.e., PointDA-10, as shown in Table 4. OOD problem is that training a model with labeled data of source domain and unlabeled data of the target domain and expecting excellent performance on the target domain. One can observe that LAMP outperforms others on average accuracy. It's noteworthy that LAMP does not have any special design for OOD recognition and only trains the network on the source domain. These results demonstrate that the frozen language-pretrained encoder in LAMP can alleviate the over-fitting on source domain, leading to better generalization performance on target domain.

Table 5: Semantic Segmentation on S3DIS Area 5.

| Method | Pre-train | mIoU (%) | mAcc (%) | Params |
|---|---|---|---|---|
| PointNet [CVPR'17] | N/A | 41.1 | 49.0 | 3.6M |
| PointNet++ [NeurIPS'17] | N/A | 53.5 | - | 1.0M |
| DeepGCN [ICCV'19] | N/A | 52.5 | - | 1.3M |
| KPConv [ICCV'19] | N/A | 67.1 | 72.8 | 15.0M |
| ASSANet [NeurIPS'21] | N/A | 66.8 | - | |
| ST [CVPR'22] | N/A | 60.0 | 68.6 | 7.8M |
| PointNext [NeurIPS'22] | N/A | 67.3 | - | 3.8M |
| Point-BERT [CVPR'22] | 3D | 60.8 | 69.9 | 21.1M |
| Pix4Point [Arxiv'22] | 2D | 67.5 | 73.7 | 21.1M |
| Image2Point [ECCV'22] | 2D | 56.6 | - | - |
| ACT [ICLR'23] | 2D | 61.2 | 71.1 | 21.1M |
| LAMP (Ours) | Language | **68.0** | **73.9** | 2.0M |

Table 6: Long-Tailed distribution on ShapeNet-Part (Yi et al., 2016) dataset.

| Model | Overall | Many | Medium | Few |
|---|---|---|---|---|
| PointNet [CVPR'17] | 80.4 | 81.9 | 91.8 | 78.1 |
| PointNet++ [NeurIPS'17] | 81.9 | 83.4 | 93.5 | 79.5 |
| DGCNN [TOG'19] | 82.3 | 83.6 | 93.5 | 81.6 |
| KPConv [ICCV'19] | 85.1 | 84.6 | **98.2** | 83.8 |
| OcCo [ICCV'21] | 83.4 | 83.9 | 95.4 | 82.5 |
| Point-BERT [CVPR'22] | 84.1 | 84.4 | 95.6 | 84.8 |
| Point-MLP [ICLR'22] | 84.6 | 84.2 | 97.6 | 83.0 |
| Point-MAE [ECCV'22] | 84.2 | 84.9 | 95.8 | 83.9 |
| P2P [NeurIPS'22] | 84.1 | 85.1 | 95.1 | 84.7 |
| PointNeXt [NeurIPS'22] | 84.2 | 85.5 | 95.7 | 83.1 |
| LAMP | **85.4** ↑ 0.3 | **85.6** ↑ 0.1 | 96.7 | **87.4** ↑ 2.7 |

Table 7: Additional Experiments on the datasets which are closer to real-world indoor scenarios.

| Model | Objaverse (%) | Shapenet-Core55 (%) | PartNet (%) | Scan200 |
|---|---|---|---|---|
| LAMP [ours] | **54.2** | **83.0** Acc | **50.5**% mIoU | **30.47** mIoU |

Moreover, we would like to highlight that compared with PointMLP (Ma et al., 2022a), LAMP outperforms advanced models like Point-MLP in terms of mean IoU scores in the dense prediction tasks of semantic segmentation and part segmentation. Meanwhile, LAMP also outperforms Point-MLP, particularly in long-tailed classes and out-of-domain recognition tasks

## 4.6 CROSS-MODAL UNDERSTANDING: 3D VISUAL GROUNDING

To further explore the advantage of LAMP, we conduct experiments on the 3D visual grounding (Achlioptas et al., 2020) task. The goal of 3D visual grounding is to locate the object in 3D scene given a text description. Referring to Table 8, we found that LAMP can directly deliver a new state-of-the-art performance on ScanRefer (Chen et al., 2020) dataset. The potential reason is that the unified encoding of LAMP mitigates the modality gap between visual and text inputs, leading to better alignment. In this case, the text feature can probe more precise visual features thanks to the improved alignment. These experimental results demonstrate that our method opens a new avenue in multimodal 3D understanding between natural language and point clouds. **Discussion on**

Table 8: **Experiment results on ScanRefer dataset.** We fairly compare LAMP with existing methods without additional 3D pretraining.

| Method | Venue | Input | Unique (%) | | Multiple (%) | | Overall (%) | |
|---|---|---|---|---|---|---|---|---|
| | | | Acc@0.25 | Acc@0.5 | Acc@0.25 | Acc@0.5 | Acc@0.25 | Acc@0.5 |
| ScanRefer (Chen et al., 2020) | ECCV'20 | 3D | 67.64 | 46.19 | 32.06 | 21.26 | 38.97 | 26.10 |
| IntanceRefer (Yuan et al., 2021b) | ICCV'21 | 3D | 77.45 | 66.83 | 31.27 | 24.77 | 40.23 | 32.93 |
| SAT (Yang et al., 2021) | ICCV'21 | 3D | 73.21 | 50.83 | 37.64 | 25.16 | 44.54 | 30.14 |
| 3DVG-Transformer (Zhao et al., 2021c) | ICCV'21 | 3D | 77.16 | 58.47 | 38.38 | 28.70 | 45.90 | 34.47 |
| 3D-SPS (Luo et al., 2022) | CVPR'22 | 3D+2D | 84.12 | 66.72 | 40.32 | 29.82 | 48.82 | 36.98 |
| 3DJCG (Cai et al., 2022) | CVPR'22 | 3D | 78.75 | 61.30 | 40.13 | 30.08 | 47.62 | 36.14 |
| BUTD-DETR (Jain et al., 2022) | ECCV'22 | 3D | 84.20 | 66.30 | 46.60 | 35.10 | 52.20 | 39.80 |
| ViL3DRel (Chen et al., 2022) | NeurIPS'22 | 3D | 81.58 | 68.62 | 40.30 | 30.71 | 47.94 | 37.73 |
| EDA (Wu et al., 2023b) | CVPR'23 | 3D | **85.76** | 68.57 | 48.11 | 36.82 | 53.83 | 41.70 |
| 3D-VisTA (Zhu et al., 2023) | ICCV'23 | 3D | 77.00 | 67.90 | 37.90 | 30.40 | 45.20 | 37.30 |
| **LAMP** | Ours | 3D | 85.62 | 69.34 | 49.83 | 38.33 | 55.17 | 42.96 |
| LAMP (Base scale + Finetuning) | Ours | 3D | 86.74 | 69.87 | 50.06 | 39.77 | 55.66 | 43.54 |
| LAMP (Large scale + Finetuning) | Ours | 3D | **88.32** | **70.75** | **51.79** | **40.23** | **57.36** | **44.60** |

**Increasing Parameters**. For multimodal tasks, with increasing trainable parameters designed in our methods, the language model backbone delivers better performance for multimodal alignments. LAMP can significantly boost joint multimodal understanding by increasing trainable parameters.

## 5 DISCUSSION AND CONCLUSION

In this paper, we propose a framework LAMP to turn traditional 3D point cloud understanding into reading a passage for language models. For multimodal research, it changes the consensus that tuning modality can be independent of pretraining modality. For 3D vision, LAMP is an effective and concise framework, which relieves the traditional hand-crafted designs on extracting representations of point clouds. Experimental results on 3D object classification and segmentation validate the superiority of the proposed LAMP framework. For many vision-language tasks, which deal with modality divergence between 3D vision and language, we believe weight-sharing across the two modalities would be a promising direction for further performance improvement.

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

# Appendix

## A  SUMMARY

### SUMMARY OF THE APPENDIX

This appendix is for the ICLR 2024 submission, titled *From Language to 3D Worlds: Adapting Language Models for Point Cloud Perception*. The appendix is organized as follows:

- § B illustrates the **intrinsic alignment** between textural and 3D representations in the high-level semantic space, where the same frozen language encoder network extracts similar representations for samples from different modalities but with the same semantic category.
- § C presents a comprehensive comparison between the proposed LAMP framework and existing approaches for point cloud understanding in terms of efficiency, including the number of training parameters, FLOPs, inference speed, *etc*
- § D provides detailed experimental settings of **long-tailed** 3D part segmentaion on the ShapeNetPart (Yi et al., 2016) dataset, and further illustrates the experimental results of 3D out-of-distribution (OOD) problem on the PointDAN dataset (Qin et al., 2019).
- § E shows the visualization results (for semantic segmentation) of the proposed LAMP and other advanced approaches on the S3DIS (Armeni et al., 2016) dataset.
- We provide the source code with detailed documentation.

## B  ALIGNMENT BETWEEN TEXT & 3D POINT CLOUDS

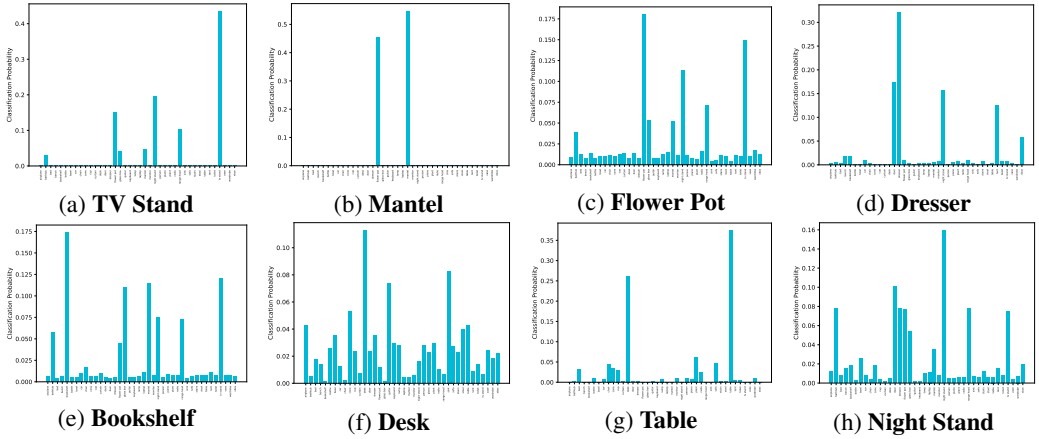

(a) **TV Stand**  (b) **Mantel**  (c) **Flower Pot**  (d) **Dresser**

(e) **Bookshelf**  (f) **Desk**  (g) **Table**  (h) **Night Stand**

Figure 4: Qualitative Results of LAMP on ModelNet-40 datasets. We visualize the semantic similarity by the dot product between the features of prompts and 3D objects, which reveals that **features of natural languages and point clouds are well aligned in the high-level semantic space**.

To uncover the underlying reason that *a language transformer can well encode the feature representations of the 3D inputs*, we measure the similarities between output features of 3D point clouds and the prompts (i.e., "A point cloud of CLS.") extracted by the same transformer encoder pre-trained on natural language (see Fig. 4). The similarity scores indicate the probability of the 3D object assigned the corresponding prompts. From Fig. 4, we can observe that the two modalities (natural language and 3D vision) are well aligned in the high-level semantic space, i.e., the features of 3D objects are matched with the corresponding prompts. This implies that the language model can leverage prompts to understand the semantics of 3D point clouds, which interprets why the language model can be exploited to encode the representations of 3D inputs.

Table 9: **Efficiency Experiment.** We report the pre-training modality (Pre-train), mAcc, OA, the number of training parameters (Train. Params.), the overall parameters (All Params.), the percentage of training parameters in all parameters (Percent.), GFLOPs, and the inference time (Infer.).

| Method | Pre-train | mAcc (%) | OA (%) | Train. Params. | All Params. | Percentt(%) | GFLOPs | Infer.(ms) |
|---|---|---|---|---|---|---|---|---|
| PointNet [CVPR'17] (Qi et al., 2017b) | N/A | 86.0 | 89.2 | 3.5M | 3.5M | 100.0 | 0.9 | 4212 |
| PointNet++ [NeurIPS'17] (Qi et al., 2017a) | N/A | - | 91.9 | 1.5M | 1.5M | 100.0 | 1.7 | 1872 |
| PointCNN [NeurIPS'18] (Li et al., 2018) | N/A | 88.1 | 92.5 | **0.6M** | **0.6M** | 100.0 | - | 44 |
| PointConv [CVPR'19] (Wu et al., 2019) | N/A | - | 92.5 | - | - | 100.0 | - | - |
| KPConv [ICCV'19] (Thomas et al., 2019) | N/A | - | 92.9 | 14.3M | 14.3M | 100.0 | - | - |
| DGCNN [TOG'19] (Wang et al., 2019b) | N/A | 90.2 | 92.9 | 1.8M | 1.8M | 100.0 | 4.8 | 263 |
| Point Transformer [ICCV'21] (Zhao et al., 2021b) | N/A | 90.6 | 93.7 | 7.8M | 7.8M | 100.0 | 5.6 | - |
| PointNeXt [NeurIPS'22](Qian et al., 2022a) | N/A | 90.8 | 93.2 | 1.4M | 1.4M | 100.0 | 1.6 | 2040 |
| Point-MLP [ICLR'22] (Ma et al., 2022b) | N/A | 90.9 | 93.6 | 0.68M | 0.68M | 100.0 | - | 176 |
| PointMixer [ECCV'22] (Choe et al., 2022) | N/A | **91.4** | 93.6 | 13.2M | 13.2M | 100.0 | - | - |
| Point-BERT [CVPR'22] (Yu et al., 2022) | 3D | - | 93.2 | 21.1M | 21.1M | 100.0 | - | - |
| Point-MAE [ECCV'22] (Pang et al., 2022) | 3D | - | 93.8 | 21.1M | 21.1M | 100.0 | - | - |
| P2P [NeurIPS'22](ResNet-101) (Wang et al., 2022) | 2D | - | 93.1 | 0.50M | 81.2M | 0.6 | - | - |
| P2P [NeurIPS'22](HorNet-L-22k-mlp) (Wang et al., 2022) | 2D | - | 94.0 | 1.2M | 205M | 0.6 | - | - |
| ACT [ICLR'23] (Dong et al., 2022) | 2D | - | 93.5 | 21.1M | 21.1M | 100.0 | - | - |
| LAMP (T5-Base (Raffel et al., 2020)) [Ours] | Language | 90.1 | 93.8 | **0.44M** | 309.4M | **0.1** | 164.2 | 242 |
| LAMP (T5-Base (Raffel et al., 2020))$_{w/}$**Vote** [Ours] | Language | 90.3 | **94.1** | **0.44M** | 309.4M | **0.1** | 164.2 | 242 |
| LAMP (T5-Small (Raffel et al., 2020)) [Ours] | Language | 89.7 | 93.4 | **0.44M** | 99.4M | 0.8 | 53.7 | 543 |
| LAMP (BERT-Base (Devlin et al., 2019)) [Ours] | Language | 88.3 | 91.9 | **0.44M** | 198.9M | 0.2 | 96.9 | 372 |
| LAMP (BERT-Medium (Devlin et al., 2019)) [Ours] | Language | 89.9 | 92.7 | **0.44M** | 22.2M | 2.0 | 19.0 | 1046 |

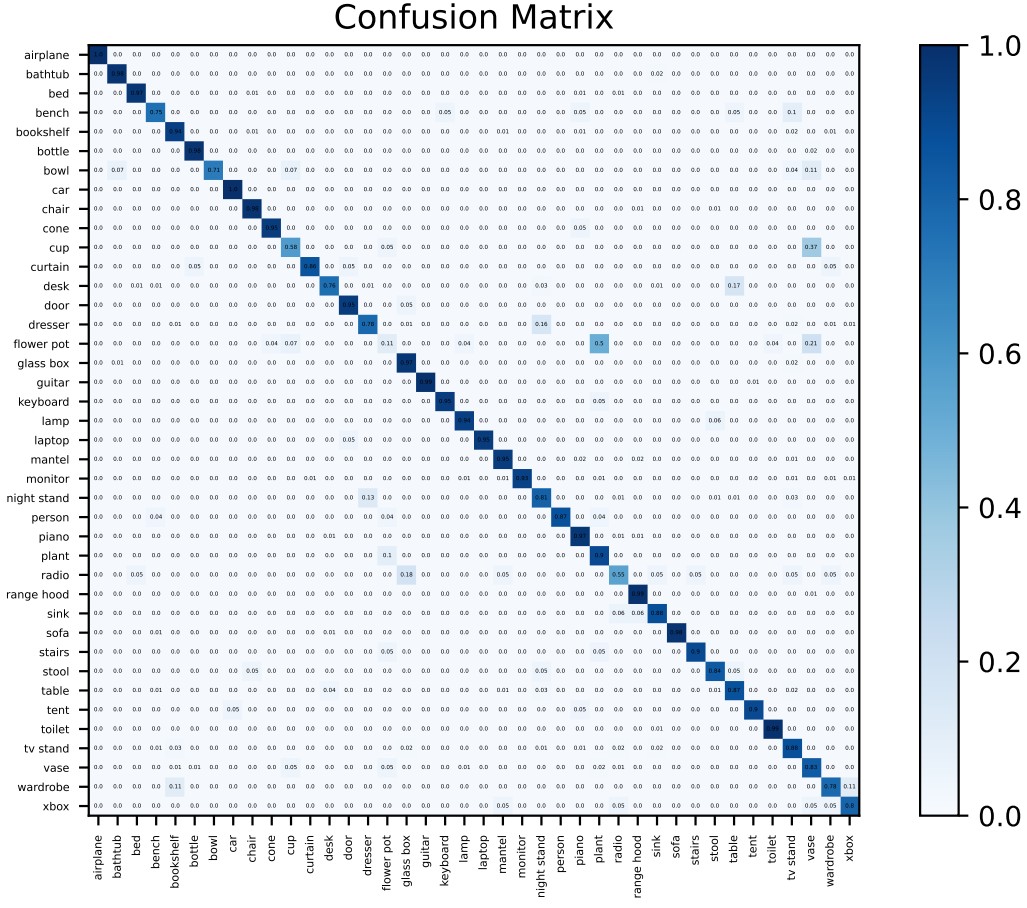

Figure 5: Confusion matrix of LAMP on the ModelNet-40 dataset. It shows that learned representations of language models for 3D point cloud can be effectively utilized for 3D object classification.

**Simiarlity Meansurement** : In specific, we utilize LAMP with a frozen BERT (Devlin et al., 2019) to extract representations of 3D point clouds $z_p$ on the ModelNet-40 dataset. For the text, the prompts are constructed as "A point cloud of CLS", such as "A point cloud of Airplane" for the airplane class. Therefore we acquire a set of prompts $\mathcal{P}$ for all classes in the ModelNet-40 dataset. We use the same BERT encoder to extract the features $z_t$ of the prompts. Particularly, we use the

output features corresponding to the [CLS] token for both the prompts and point clouds. With the $z_p$ and $z_t$ at hand, the dot product is utilized to estimate the similarities between the 3D objects and the prompts, followed by a softmax function to calculate the probabilistic scores. Formally, the similarities scores can be formulated as:

$$s_{i,j} = \frac{\exp(z_p^i \cdot z_t^j / \tau)}{\sum_{k \in \|\mathcal{P}\|} \exp(z_p^i \cdot z_t^k / \tau)}, \tag{5}$$

where $\tau$ is temperature factor as (He et al., 2020) and $s_{ij}$ indicates the similarity scores of $i^{th}$ point cloud and $j^{th}$ prompt.

Furthermore, we visualize the confusion matrix of the proposed LAMP as shown in Fig. 5, from which one can observe that the LAMP model can recognize each object class with minimal confusion.

**Discussion between light-weight Models** Compared with representative light-weight methods such as RepSurf (Ran et al., 2022), LAMP's superior performances, **particularly in dense prediction tasks and under challenging conditions like long-tailed and out-of-distribution problems**.

| Model | ModelNet (%) | S3DIS (%) | ShapeNetPart (%) | Long-Tail (%) | PointDA-10 (%) |
|---|---|---|---|---|---|
| RepSurf | 94.7 OA | 68.9 mIoU | 85.3 mIoU | 83.6 mIoU (Few) | 71.3 Acc. |
| LAMP | 93.8 OA | 68.0 mIoU | 86.9 mIoU | 87.4 mIoU (Few) | 74.0 Acc. |

Table 10: Performance comparison of light-weight models on various datasets.

## C EFFICIENCY EXPERIMENTS

Beyond the empirical evaluation using typical metrics, we perform auxiliary performance on efficiency w.r.t. the number/percentage of training parameters, FLOPs, and the inference speed (the consuming time for all test samples), to validate the merit of our LAMP, as shown in Table 9.

Compared with other competitive methods, our prosed LAMP yields the best performance of 93.8% OA while only training 0.44M of network parameters and consuming 242 seconds for the testing stage. It's also noteworthy that LAMP only requires training 0.1% of total parameters, 6× less than its image counterpart, P2P. A smaller cost is needed to adapt the language model to 3D vision domain while greater performance can be achieved than the image models, which reveals the inherent advantage of language in terms of processing 3D point clouds.

## D LONG-TAIL AND OOD PROBLEMS IN 3D VISION

### D.1 LONG-TAIL SEGMENTATION

In total, there are 16, 880 objects of 16 different shape categories in the ShapeNetPart dataset (Yi et al., 2016). The detailed data distribution is presented in Table 11. We split the datasets based on the data number of classes. For example, the classes with more samples, such as "Table" and "Car", are cast as the **Many** classes while the rest is cast as **Medium** or **Few**.

### D.2 OUT-OF-DISTRIBUTION IN 3D VISION

Typically, supervised algorithms only focus on the in-domain performance, where the training and testing data are sampled from an identical distribution. However, in practice, the model is often exposed to the out-of-domain (OOD) scenario, i.d., the test data severely differ from the training one. Therefore, we aim to evaluate whether the language model can help the point cloud models to generalize well on the out domains. To this aim, we first fine-tune our input and output layers of LAMP on a training domain and then directly test it performed on a different domain without any extra adaptation training stage like domain adaptation approaches (Pan & Yang, 2010; Long et al.,

Table 11: Data distribution on the ShapeNetPart dataset (Yi et al., 2016), where N denotes the number of objects, L denotes the number of parts for objects, and Ann denotes the annotations.

| Type | Category | N | L | Ann |
|---|---|---|---|---|
| Many | Table | 8420 | 2 | 999 |
| | Car | 7496 | 3 | 2215 |
| | Chair | 6742 | 4 | 2112 |
| | Airplane | 4027 | 4 | 2142 |
| | Lamp | 2308 | 3 | 696 |
| Medium | Guitar | 793 | 3 | 270 |
| | Laptop | 452 | 1 | 18 |
| | Knife | 420 | 2 | 149 |
| | Motorbike | 336 | 5 | 264 |
| | Pistol | 307 | 3 | 176 |
| | Mug | 213 | 1 | 33 |
| | Skateboard | 152 | 2 | 24 |
| | Rocket | 85 | 3 | 35 |
| Few | Bag | 83 | 2 | 18 |
| | Earphone | 73 | 2 | 25 |
| | Cap | 56 | 2 | 18 |
| Total | All | 31963 | 42 | 9194 |

2015). The OOD experiment results are shown in Table 4, from which we can observe that our proposed LAMP reaches encouraging performance on the unseen domains, yielding the best average performance of 74.0%. Even compared with the domain adaptation methods like DANN (Shen et al., 2022), PointDAN (Qin et al., 2019), and the recent Implicit PCDA (Shen et al., 2022), LAMP can still outperform them in term of the average performance. The potential interpretation is that the supervision signals from the source domain might cause over-fitting, thus limiting the generalization performance. In addition, the results reveal that prior knowledge from natural language is helpful to mitigate the over-fitting of the training domain, and has a better generalization ability to new domains.

## E    VISUALIZATION RESULTS

We illustrate the extra visualization results on the S3DIS (Armeni et al., 2016) dataset of our LAMP and other approaches including PointNet (Qi et al., 2017b), Point-BERT (Yu et al., 2022), Point-NeXt (Qian et al., 2022a) as shown in Fig. 6. One can observe that LAMP can yield better segmentation results than others.

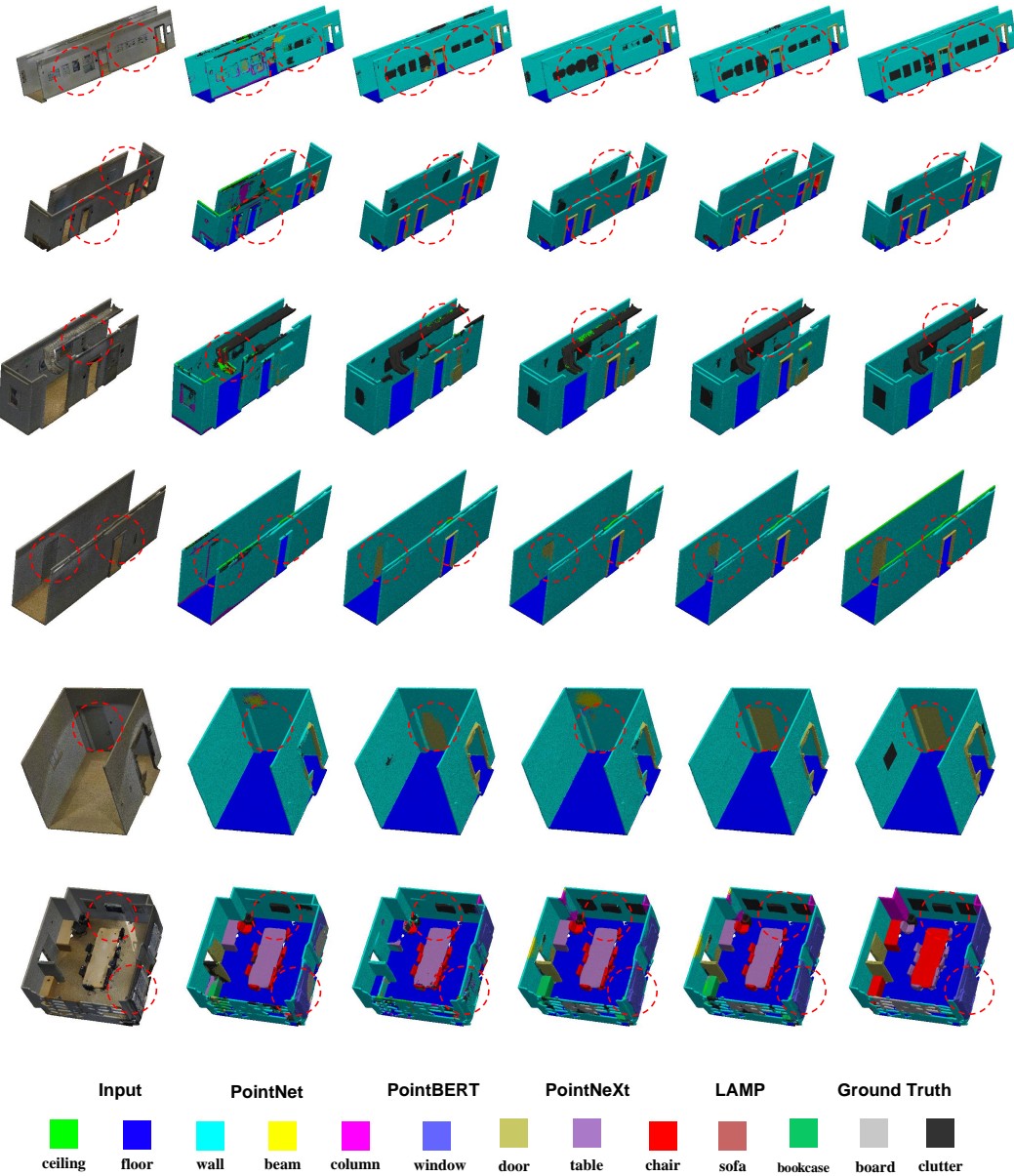

Figure 6: Qualitative Results of LAMP on S3DIS Area 5. LAMP with natural language corpus pre-training (5th column) achieves better segmentation results than advanced 3D understanding methods including PointNet (Qi et al., 2017b) (2nd column), Point-BERT (Yu et al., 2022) (3rd column) and PointNeXt (Qian et al., 2022a) (4th column).

