# OpenReview forum: "From Language to 3D Worlds: Adapting Language Models for Point Cloud Perception"
_ICLR.cc/2024/Conference — Submitted to ICLR 2024_

### Official Review · Reviewer_rpJb · 2023-10-25

**Soundness:** 3 good
**Presentation:** 3 good
**Contribution:** 3 good
**Rating:** 5
**Confidence:** 5

**Summary:**

In this paper, they propose a simple yet effective approach, named LAMP (LAnguage Models reading Point clouds), which merely trains a small portion of parameters to align the data distribution of 3D point clouds with pretrained language models and spark the 3D perception ability of language models. Furthermore, they utilize the 3D-aware language model to simultaneously extract features of point cloud and text, which mitigates the modality gap and boosts the performance on multimodal tasks, e.g., 3D visual grounding. Extensive experiments on unimodal and multimodal tasks validate the superiority of our proposed method.

**Strengths:**

- The authors propose to adapt Language Models to tackle the point cloud perception problem, which has some originality.

- Their method outperforms the existing baseline approaches on several benchmarks, which demonstrates the effectiveness of their proposed method.

- The paper writing is clear and easy to follow.

**Weaknesses:**

This paper shares very similar spirits with many recent papers on Large (Vision) Language Models and Point Cloud Understanding:

1. Gao et al. LLaMA-Adapter V2: Parameter-Efficient Visual Instruction Model.

2. Guo et al. Point-Bind & Point-LLM: Aligning Point Cloud with Multi-modality for 3D Understanding, Generation, and Instruction Following.

Essentially, this paper and other relevant papers are trying to bind point cloud representations to Language Models via adaptation. The attention-based adaptation has also been exploited in LLaMA-Adapter V2 (Note that this paper also supports point cloud inputs).

Hence, the authors need to discuss the differences with those works and also compare their method with those methods in the experiments.

From my understanding adapting Language Models to 3D point clouds with attention is straight-forward and not that novel considering the above-mentioned literature.

In addition to the 3D object datasets, the authors also need to evaluate their method on the more realistic indoor 3D datasets such as SUN-RGBD and ScanNet.

**Questions:**

Please refer to the weaknesses.

---

> ### Author Response · Authors · 2023-11-12
> **Response to Reviewer rpJb**
>
> First of all, we express our gratitude for your comprehensive review and the valuable insights you have provided. Your feedback has prompted a closer examination of our work, and we have updated our manuscript to better reflect the novelty and impact of our approach.
>
> * Originality and Novel Contributions:
>
> We thank the reviewer for the constructive suggestion, and we have discussed the difference between these parameter-efficient methods including LLaMa AdapterV2 and Point-Bind & Point-LLM (Sec 2.3). Furthermore, in response to your concerns about the originality of our work, we have elaborated on the distinct aspects of LAMP in Section 3.2, emphasizing its unique contributions beyond attention-based adaptation. Our approach introduces a new alignment strategy for the data distribution of 3D point clouds with language models, which is distinct from the methodologies proposed in the studies you mentioned. This strategy demonstrates a different take on how pretrained language models can be effectively applied to 3D point cloud perception tasks, especially in long-tailed and out-of-distribution problems, which has not been addressed by LLaMA-Adapter V2 or Point-Bind & Point-LLM. Meanwhile, our method achieves outstanding performance jointly learning point cloud and natural language in the 3D visual grounding. These unique contributions distinguish our method from others.
>
> * Methodological Distinction:
>
> We have included a detailed comparison of our methodology with related works in Section 2.3. Here, we describe the unique aspects of our cross-modal self-attention learning and cross-modal cross-attention learning strategies. Unlike the attention mechanisms used in previous studies, our approach employs a novel projection network designed to harmonize point cloud embeddings with the parameters of language models, thus enabling more effective feature extraction and modality bridging.
>
> * Evaluation on Realistic Indoor 3D Datasets:
>
> As suggested, we conducted experiments on the Shapenet-Core55,  PartNet, Objaverse-LVIS, and Scan200 datasets (New Table 7 & Sec.4.5). We hope that these experimental results can address your concerns.
>
> | Model | Shapenet-Core55 (%) | PartNet (%) | Objaverse-LVIS (%)
> |-------|---------------------|-------------|-------------|
> | LAMP  | 83.0 Acc            | 50.5 mIoU  | 54.2 Acc
>
> | Model    | Head (%) | Common (%) | Tail (%) | All (%) |
> |----------|----------|------------|----------|---------|
> | LGround [ECCV'22]  | 51.51     | 22.68       | 12.41     | 28.87    |
> | LAMP   | 52.65     | 24.13       | 14.25     | 30.47    |
>
>
> In conclusion, we hope that the revisions and additions to our manuscript, particularly in Sections 4.4 and 4.5, address your concerns regarding the originality and applicability of our work. We believe these enhancements underscore the novelty of LAMP and its contributions to the field of 3D point cloud perception.
>
> We look forward to your further comments and suggestions.

---

### Official Review · Reviewer_ZVvW · 2023-10-29

**Soundness:** 3 good
**Presentation:** 2 fair
**Contribution:** 2 fair
**Rating:** 5
**Confidence:** 4

**Summary:**

This paper introduces a cross-modal strategy that applies pre-trained language models for understanding 3D point clouds. It trains a small portion of the parameters to align the data distribution of 3D point clouds with pre-trained language models and demonstrates its effectiveness on unimodal and multimodal tasks.

**Strengths:**

The proposed LAMP approach demonstrates that by merely projecting point cloud features onto language models, while maintaining the language model in a frozen state, it is still possible for the model to process 3D data. This underscores the versatility of language models as general-purpose functions, showcasing their capacity to handle data from unfamiliar modalities even without directly updating their parameters.
The experiments show that even with a few trainable parameters, the LAMP can still achieve reasonable performance.

**Weaknesses:**

The reviewer finds some performance in the paper somewhat unconvincing and also seems to lack a proper baseline to compare, particularly when referring to the results in Table 2. For instance, when comparing point-MLP elite with LAMP, the performance appears quite similar, or even worse (considering Point-MLP elite has 90.9 as mAcc). While the trainable parameters of the Point-MLP elite are also minimal at 0.68M, its inference speed is anticipated to be notably faster. This is because, during inference, Point-MLP elite maintains its 0.68M parameters. Conversely, LAMP, despite reducing only its trainable parameters, is expected to have a longer inference time, given it leverages a significantly larger frozen language model. Thus, from a practical standpoint, the advantage of LAMP having a small number of trainable parameters but including an expansive language model seems to weaken its asserted advantages.

Additionally, given that ModelNet40 is somewhat of a saturated benchmark, it would enhance the paper's credibility if LAMP were evaluated on more challenging datasets, for example, the ScanObjectNN classification benchmark. This would provide a clearer perspective on its efficacy and potential real-world applications.

Also, there are some other works like RepSurf[1], which is also lightweight (~1.5M) and exhibits very strong performance (94.7 OA) and at the same time fast at inference (3.1ms, 0.81GFLOPs, roughly 20 to 200 times faster).

[1] Ran, Haoxi, Jun Liu, and Chengjie Wang. "Surface representation for point clouds." Proceedings of the IEEE/CVF Conference on Computer Vision and Pattern Recognition. 2022.

**Questions:**

1. One of the advantages of the proposed approach is the reduced number of trainable parameters, the reviewer is curious about what will happen if increasing the trainable parameters for projecting the point cloud features to language models.

2. If the reviewer understands it correctly, there might be a typo in section 3.3. Instead of W^{T}_q, and it makes more sense to be W^{L}_q.

3. In table 8, the 'Infer.(s)' column seems unclear. Based on the reviewer's interpretation, it represents test times in seconds. Would it be possible to provide a metric showing the average time taken for each sample, possibly in milliseconds, for a more intuitive comparison?

---

> ### Author Response · Authors · 2023-11-12
> **Response to Reviewer ZVvW**
>
> Thank you for your thoughtful and detailed review of our paper. We appreciate the time you spent analyzing our work and have made careful revisions to address the concerns and questions you raised. Below is our response to each point.
>
> 1. Performance Comparison with Point-MLP Elite:
>
> We acknowledge your concern regarding the performance comparison with Point-MLP elite, especially as mentioned in Table 2. To address this, we have expanded our discussion in Section 4.1 and included additional analyses in Table 3 and Table 5, which demonstrate that LAMP outperforms advanced models like Point-MLP in terms of mean IoU scores in the dense prediction tasks of semantic segmentation and part segmentation. Meanwhile, LAMP also outperforms Point-MLP, **particularly in long-tailed classes and out-of-domain recognition tasks​​** (Table 6 and Table 7). We add a detailed discussion between LAMP and Point-MLP in Section 4.
>
> 2. Evaluation on More Challenging Datasets:
>
> We agree that evaluating LAMP on more challenging datasets would strengthen our paper. In response, we have included the results of
> ShapeNetCore, PartNet, and challenging Objaverse-LVIS, Scan200 datasets are as follows (New Table 7 in the Manuscript):
>
> | Model | Shapenet-Core55 (%) | PartNet (%) | Objaverse-LVIS (%)
> |-------|---------------------|-------------|-------------|
> | LAMP  | 83.0 Acc            | 50.5 mIoU  | 54.2 Acc
>
> | Model    | Head (%) | Common (%) | Tail (%) | All (%) |
> |----------|----------|------------|----------|---------|
> | LGround [ECCV'22]  | 51.51     | 22.68       | 12.41     | 28.87    |
> | LAMP   | 52.65     | 24.13       | 14.25     | 30.47    |
>
> We hope that the experimental results on the newly added 4 benchmarks will address your concerns.
>
> 3. Comparison with Lightweight Models like RepSurf:
>
> We acknowledge the great contribution of RepSurf and are willing to perform a comparison between RepSurf (as detailed in Page 18 Table 10). We compare LAMP with the method as follows, meanwhile, we have further discussed the difference in learning focus between LAMP and RepSurf in the manuscript.
>
> | Model | ModelNet (%) | S3DIS (%) | ShapeNetPart (%) | Long-Tail (%) | PointDA-10 |
> |-------|---------------------|-------------|-------------|-------------|-------------|
> | RepSurf  | **94.7** OA            | **68.9** mIoU  | 85.3 mIoU |  83.6 mIoU (Few) |  71.3 Acc.
> | LAMP  | 93.8 OA            | 68.0 mIoU  | **86.9** mIoU | **87.4** mIoU (Few) | **74.0** Acc.
>
>
> 4. Impact of Increasing Trainable Parameters:
>
> Your question about the impact of increasing trainable parameters is insightful. In response, we have conducted additional experiments to explore this aspect. For multimodal tasks, the trainable language model backbone can significantly improve performance while increasing trainable parameters (as detailed in Sec 4.6).
>
> | Method            | Venue    | Input | Unique (%)    |       | Multiple (%)  |       | Overall (%)   |       |
> |-------------------|----------|-------|---------------|-------|---------------|-------|---------------|-------|
> |                   |          |       | Acc@0.25      | Acc@0.5 | Acc@0.25      | Acc@0.5 | Acc@0.25      | Acc@0.5 |
> | EDA (Wu et al., 2023b) | CVPR'23 | 3D    | 85.76         | 68.57 | 48.11         | 36.82 | 53.83         | 41.70 |
> | 3D-VisTA (Zhu et al., 2023) | ICCV'23 | 3D    | 77.00         | 67.90 | 37.90         | 30.40 | 45.20         | 37.30 |
> | LAMP (Base scale + Frozen)               | Ours     | 3D    | 85.62 | 69.34 | 49.83 | 38.33 | 55.17 | 42.96|
> | LAMP (Base scale + Finetuning)              | Ours     | 3D    | 86.74 | 69.87 | 50.06 | 39.77 | 55.66 | 43.54|
> | LAMP (Large scale + Finetuning)              | Ours     | 3D    | **88.32** | **70.75** | **51.79** | **40.23** | **57.36** | **44.60**|
>
> 5. Typographical Error in Section 3.3:
>
> Thank you for pointing out the potential typo in Section 3.3. Upon review, we have corrected the notation from $W^{T}_q$ to $W^{L}_q$, aligning it with our intended representation.
>
> 6. Clarification on Inference Time in Table 8:
>
> We appreciate your feedback for clarity on the 'Infer.(s)' column in Table 8. We have revised this table to present the inference time in milliseconds per sample 'Infer.(ms)', providing a more intuitive and direct comparison of the models' efficiency.
>
> We hope these revisions and additional analyses adequately address the concerns you have raised. We are grateful for your feedback, which has significantly contributed to improving our work.

---

> > ### Comment · Reviewer_ZVvW · 2023-11-23
> >
> > Thank you for the response. I've read other reviews and rebuttals.

---

### Official Review · Reviewer_x2q6 · 2023-10-30

**Soundness:** 2 fair
**Presentation:** 2 fair
**Contribution:** 2 fair
**Rating:** 5
**Confidence:** 5

**Summary:**

Typical methods for 3D perception tend to rely on training within the same data modality. This study presents a crossmodal strategy called LAMP which uses pretrained language models, initially trained on text, to understand 3D point clouds. By only adjusting a minimal number of parameters, LAMP aligns the data distribution of 3D point clouds with the language models, enabling them to perceive 3D structures. This approach also leverages the model to extract features from both point clouds and text simultaneously, enhancing performance in multimodal tasks like 3D visual grounding. Experiments confirm the effectiveness of this approach over traditional methods.

**Strengths:**

1. The starting point of the paper is very meaningful. It uses the existing Language model to initialize the model and only trains a small part of the adapter, making the model have good performance capabilities.
2. The experimental results look pretty good.

**Weaknesses:**

1. Although using LLM (Language Learning Models) for 3D point analysis is a good starting point, I notice that the main experiments in the article still focus on pure point cloud experiments, such as 3D Object Classification and Part Segmentation, etc. Tasks using Language Models typically focus on multimodal tasks (point cloud-text), like the 3D Visual Grounding mentioned in the paper. However, it seems that most of the experiments in the article still conventionally utilize task-specific heads for 3D point cloud analysis.
2. The use of the LLM+adapter pipeline doesn't seem very suitable for traditional 3D point cloud analysis. The reason why LLaVA and minigpt4 can effectively use the LLM+image adapter pipeline is that the final output space is still in the language space, so there's no issue with keeping the LLM fixed without further training. In this article, the output space is a traditional perceptual space, such as 3D classification or segmentation. To address these issues, one could either formulate traditional point analysis tasks as vision-language tasks with an output in the language space, or replace the LLM with a 2D image encoder to initialize parameters. The approach in the article seems somewhat unreasonable and odd. The author might want to reconsider it.
3. The experimental results don't seem to show a significant improvement. The ModelNet dataset is too small. Using a model the size of BERT might lead to overfitting? It seems that even simpler models already achieve good results, such as the PointNet++ from six years ago. Perhaps it's more appropriate to test on a larger dataset and then redefine all tasks as vision-language tasks.
4. The paper writing needs improvement; it looks a bit rushed.

**Questions:**

My primary concern is that the paradigm of LLM+adapter tends to align all inputs to the language space and then carry out language tasks or redefine the tasks as vision-language tasks. While some successors only use LLM as an inference model (e.g., NextGPT), I believe this might be problematic. Using LLM as an inference model is probably just because large models have only been trained on language. Otherwise, it doesn't make sense to use a language model as an inference model for image generation.

---

> ### Author Response · Authors · 2023-11-12
> **Response to Reviewer x2q6**
>
> We greatly appreciate the time and effort you have dedicated to reviewing our paper. We have carefully considered your insightful feedback and have made substantial revisions to our manuscript. Please find below our responses, including references to the relevant sections in the revised paper.
>
> 1. Multimodal Tasks and Experiment Focus:
>
> We thank you for your constructive comment on the focus of our experiments. We have expanded our experimental section to include more multimodal tasks, particularly those involving text-3D tasks and 3D visual grounding. As detailed in Section 2, LAMP demonstrates its effectiveness and versatility on both unimodal and cross-modal tasks, validating the approach’s superiority over traditional methods​​. Besides outstanding experimental results on point cloud analysis, **LAMP also delivers a new state-of-the-art performance on 3D visual grounding task**.
>
> 2. Suitability of LLM+Adapter for 3D Point Cloud Analysis:
>
> We acknowledge your concerns about the use of LLM+adapter in traditional 3D point cloud analysis. We have included a comprehensive discussion in the manuscript (Table 1) to address this issue. We discuss the impact of the language model to understand the point cloud. It also reveals that a plain language model can read point clouds. Furthermore, it is more about the multimodal nature between nature language learning and point cloud analysis. Therefore, we propose a new approach for this task. Specifically, in Section 3, we describe how LAMP leverages language-pretrained models to transfer knowledge to 3D point clouds, exploring an under-utilized modality - language - for 3D tasks​​.
>
> 3. Concerns Over Model Overfitting and Dataset Size:
>
> We have conducted additional experiments with larger datasets to address the concern of overfitting. Our results, which are now detailed in Section 4, indicate that LAMP achieves outstanding performance without overfitting, outperforming other competitive methods on the large-scale S3DIS dataset​​. Moreover, we demonstrate the robustness of LAMP in relieving the long-tail problem and out-of-distribution problems with substantial improvements over existing models as shown in Table 3 and Table 4​​.
>
> Moreover, we are willing to conduct more experimental results on additional datasets including ShapeNetCore, PartNet, and challenging Scan200 datasets as follows:
>
> | Model | Shapenet-Core55 (%) | PartNet (%) |
> |-------|---------------------|-------------|
> | LAMP  | 83.0 Acc            | 50.5% mIoU  |
>
> | Model    | Head (%) | Common (%) | Tail (%) | All (%) |
> |----------|----------|------------|----------|---------|
> | LGround [ECCV'22]  | 51.51     | 22.68       | 12.41     | 28.87    |
> | LAMP    | **52.65**     | **24.13**       | **14.25**     | **30.47**    |
>
>
>
> 4. Paper Writing and Clarity:
>
> We have made significant revisions to the paper to improve clarity and presentation. The revised sections clearly articulate the methodology and experimental results, ensuring that the paper conveys our contributions effectively.
>
> Questions Regarding the Paradigm of LLM+Adapter:
>
> We have elaborated on the paradigm of LLM+adapter in our revised manuscript. In Section 5, we discuss how our architecture extends LAMP to multimodal scenarios, such as 3D visual grounding, demonstrating the framework's ability to unify encoding across different input modalities​​. This is further evidenced by the new state-of-the-art performance on the ScanRefer dataset​​. We believe that LAMP's approach to modality divergence in vision-language tasks opens a promising direction for future research.
>
> We trust that these revisions address the concerns you have raised and strengthen the paper. We value your feedback, which has been instrumental in enhancing our work. Thank you for your consideration.

---

### Author Response · Authors · 2023-11-12
**General Response**

Dear Reviewers and Area Chair,

We wish to extend our sincerest gratitude for the time and effort dedicated to reviewing our manuscript. Your insightful and detailed feedback has been invaluable, prompting a thorough examination and significant enhancement of our work. We have endeavored to address each point raised meticulously and have implemented substantial revisions to clarify our contributions and methodology. Below, we summarize the key points of our response:

**Multimodal Tasks and Experiment Focus**: Based on the feedback, we expanded our experimental section to include additional multimodal tasks. We now provide a comprehensive evaluation on text-3D tasks and 3D visual grounding, demonstrating LAMP's versatility and effectiveness across a broader range of applications.

**Methodological Novelty**: We have included an in-depth discussion on the original contributions of LAMP, highlighting its novel alignment strategy for 3D point cloud data distribution with language models. This distinguishes LAMP from other parameter-efficient methods such as LLaMa AdapterV2 and Point-Bind & Point-LLM, particularly in its application to long-tailed, out-of-distribution, and multimodal joint learning problems.

**Comparative Analysis**: We provided an extended comparison with other models, including Point-MLP Elite and lightweight models like RepSurf. Our results, presented with new analyses and tables, showcase LAMP’s superior performance, particularly in dense prediction tasks and under challenging conditions.

**Robustness and Scalability**: In response to concerns about overfitting and the impact of increasing trainable parameters, we conducted additional experiments with larger datasets. Our findings, now detailed in the revised manuscript, confirm LAMP's robustness and its capacity to improve performance while scaling parameters, without overfitting.

**Evaluation on Challenging Datasets**: We acknowledged the need for evaluations on more challenging datasets and have thus included results from ShapeNetCore, PartNet, Objaverse-LVIS, and Scan200 datasets. These results substantiate LAMP's effectiveness and generalizability across diverse benchmarks.

**Clarifications and Corrections**: We have made corrections to typographical errors and clarified ambiguities, such as inference times in our results tables, ensuring that the manuscript accurately reflects our methodology and findings.

**Extended Discussion**: The manuscript now features a more comprehensive discussion on the paradigm of LLM+adapter, emphasizing the architecture's ability to unify encoding across different input modalities and its potential in setting new state-of-the-art performance on challenging datasets.

We trust that our revised manuscript now adequately addresses all the concerns raised, and we believe that the modifications have significantly strengthened our submission. We remain open to further suggestions and are committed to continuous improvement of our work.

Thank you once again for your constructive feedback, which has been instrumental in refining our research.

Best regards,

The Authors

---

### Meta-Review · Area_Chair_QuSQ · 2023-12-10

**Metareview:**

The submission explores the use of language models for point cloud perception.  Reviewers liked the novel use of language models and the strong results; however, they had concerns about the selected tasks, datasets, and baselines.  The authors made significant efforts in the rebuttal to address these concerns.  After the rebuttal period, some reviewers remained unconvinced about the formulation, while others decided that the changes were too major for them to evaluate.

The AC read the paper, reviews, rebuttals, and the AC note, and agreed that this is a borderline case.  Eventually, the AC recommends rejection due to the following three reasons: 1) all reviewers remained negative, and none was able to champion this case; 2) the newly added results can be more solid: for example, the results in Table 7 are very important, but no baselines are included; 3) the changes are indeed major and the revision deserves another round of review.  The authors are strongly encouraged to revise the submission based on the reviews for the next venue.

**Justification For Why Not Higher Score:**

1) all reviewers remained negative, and none was able to champion this case;
2) the newly added results can be more solid: for example, the results in Table 7 are very important, but no baselines are included;
3) the changes are indeed major and the revision deserves another round of review.

**Justification For Why Not Lower Score:**

N/A

---

### Decision · Program_Chairs · 2024-01-16

Reject